# Determinants of the happiness of adolescents: A leisure perspective

**Eui-Jae Kim[1], Hyun-Wook Kang[1]\*, Seong-Man Park[2]\***

**1** Department of Recreation and Leisure Sports, College of Sport Science, Dankook University, Cheonan, Chungcheongnam-do, Republic of Korea, **2** Department of English Language, College of Foreign Languages, Dankook University, Cheonan, Chungcheongnam-do, Republic of Korea

\* leon5989@hanmail.net (H-WK); seongmanpark@dankook.ac.kr (S-MP)

## Abstract

Leisure plays a key role in the happiness of youth. Studies have shown that various factors of leisure, such as the type, the time, the cost, and the space, have an influence on the adolescents' happiness. However, little is known about which of these factors is a major factor in youth's happiness. The purpose of this study is to explore the leisure factors that determine happiness in adolescents by examining the relationship between happiness and various leisure factors. The study used the method of machine learning to analyze national statistical data, National Leisure Activity Survey. The data used in this study were from the National Leisure Activity Survey 2019, which is a national statistic produced by the Ministry of Culture, Sports and Tourism in the Republic of Korea. The analysis found that leisure perceptions, academic and leisure balance, and public leisure space have a very important impact on the adolescents' well-being. The findings of this research may contribute to a better understanding of leisure and happiness in adolescents, and will also help adolescents make better use of their leisure time, leading to better leisure lives, and ultimately contribute to raising their level of happiness.

**Data Availability Statement:** All relevant data are within the manuscript and its Supporting Information files.

**Funding:** The author(s) received no specific funding for this work.

## Introduction

Leisure plays an important role in the enhancement of the quality of our lives. Leisure activities experienced in childhood are particularly significant in that they can develop into leisure life in adulthood [1–4]. Therefore, it is vital that the youth make good use of leisure time and lead wholesome leisure lives.

There is a lot of literature that shows the effect of participation in leisure activities on adolescents. It is found that participation in leisure activities in adolescence is associated with lowered levels of antisocial behavior, violence, substance use, and depression [5–9]. Other studies have found that there is a positive relationship between leisure participation and academic achievement, civic engagement, and mental health [10–12]. Leisure activities in adolescence can ultimately be linked to better happiness [13].

The link between leisure and happiness in adolescents has been proven through many empirical studies. Many researchers have focused on the mode of participation in leisure

**Competing interests:** The authors have declared that no competing interests exist.

activities, such as variety, type, companionship, and frequency of leisure activities [14–19] and have found that leisure resources such as leisure time, cost, space, and facilities have an effect on happiness [20–24]. Other studies have found that the perceptions and attitudes that the participants have toward leisure, and the balance between study and leisure, affect happiness [25–29].

A comprehensive review of studies on adolescent leisure and happiness show that adolescent happiness is influenced by a variety of leisure factors. However, it is difficult to find information on which of these factors are crucial to a youth's happiness. Existing studies have focused on linking happiness to a leisure factor and interpreting it simply as a unitary. Some studies analyze the relationship between multiple leisure factors and happiness, but this also has limitations that fail to take into account various leisure factors. Understanding the complex relationship between leisure and happiness in adolescents require multidimensional studies that take into account many varieties of leisure factors.

Therefore, this study seeks to empirically analyze the relationship between various leisure factors and happiness. Through this, the main leisure factors that affect the happiness of adolescents will be explored. This study will help adolescents make good use of leisure time, lead a good leisure life, and ultimately contribute to raising their level of happiness. Furthermore, the findings of this study will be able to contribute to the increase of awareness about the relationship between leisure factors and happiness and eventually both to mental and physical health related features of sustainability among adolescents in the Republic of Korea.

## Literature review

### Concept and determinants of happiness

The universal yet ultimate goal pursued by humans is to lead a happy life [30]. Happiness is a very abstract term among various psychological terms. Therefore, the definition of happiness is somewhat different among scholars. Until now, happiness has been used without much distinction from concepts such as quality of life, subjective well-being, and life satisfaction. There was an opinion that there is no practical benefit to classifying these concepts because they have a high correlation with each other [31]. On the other hand, some studies explain these concepts separately [32].

First of all, quality of life is the concept most compared to happiness. Quality of life is defined as a concept that includes objective well-being and subjective well-being. Diener (1984) [33] defined happiness as subjective well-being and argued that happiness is determined by subjective judgment about one's life. In the case of subjective well-being, it can be said to be distinguished from other similar concepts of happiness in that it emphasizes immediate satisfaction rather than overall satisfaction in life. On the other hand, some researchers explain happiness in relation to life satisfaction. Veenhoven (2010) [34] defined happiness as "subjective appreciation of life" (p. 608). Kalmijn and Arends (2010) [35] explained happiness as "the subjective appreciation of one's life-as-a-whole" (148). To sum up, the definition of happiness can be viewed from various perspectives, but it can be seen as a comprehensive concept that goes beyond pleasure and includes the emotional state of an individual and the cognitive evaluation of one's life.

Then, what is it that determines happiness? This question has long been a major concern for many researchers. Income is the area that has been most actively discussed in explaining happiness [36,37]. In addition, health [38,39], education [40,41], labor [42–44], Family [45,46], and politics [47,48] have been found to be determinants of happiness. Furthermore, leisure is an important factor in determining happiness [49–51].

## Factors that influence the happiness of adolescents

One of the factors that influence the happiness of adolescents is the diversity of leisure activities. Dahan-Oliel, Shikako-Thomas, and Majnemer (2012) [52] found in a systematic review that participation in varieties of leisure activities was associated with improved quality of life (QoL). Santini et al. (2020) [17] found that adolescents who participate in different types of leisure activities have better mental health than those who engage in only one type of leisure activity. In addition, Roman Kralik (2023) [53] argued that leisure activities in a variety of environments are helpful to children, the younger generations, and further to the society as a whole, and thus, the free time of young people should be used in a desirable way to positively affect the lives of the young people outside of school. In a similar vein, lack of outdoor activities and interactions offline due to the increased use of the Internet along with the implementation of online classes during the COVID-19 pandemic caused lack of well-being, loss of activity and vitality, and an increase of stress among students and the younger generations of recent days, which might imply the importance of physical activities and interactions offline for the young people [54].

The type of leisure activities is one of the most studied factors affecting adolescents' happiness. Many studies emphasize the importance of active leisure activities such as physical activity and social activity [14,18,55–57]. In particular, reportedly volunteer work has shown to have an important relationship with well-being and health [58–62]. In addition to active and social leisure activities, studies have also highlighted the importance of passive leisure activities such as watching TV, listening to music, and reading [63,64]. A notable aspect of passive leisure activities is the use of digital devices. It is argued that the use of digital media has a positive effect on the well-being of adolescents [65,66]. On the other hand, some studies have shown that digital media use may not necessarily be positive [19,67].

There is evidence that companionship also affects the happiness of youth. Spending leisure time with family or friends is associated with higher levels of happiness than spending leisure time alone [16,68,69]. However, spending your free time with others is also not necessarily absolute. Larson and Csikszentmihalyi (2014) [70] found that a moderate amount of alone time has a positive effect on adolescents.

Depending on the frequency of leisure activities, adolescents' levels of happiness also may vary. Adolescents who participate in leisure activities more frequently have higher levels of happiness than those who are less likely to participate in leisure activities [15,71]. On the other hand, some studies have shown that adolescents with a higher frequency of participation have lower levels of happiness [72], and studies have shown that it is not the frequency of participation in itself that determines the level of happiness in adolescents [73].

Leisure resources such as leisure time, money, and space (places, facilities) also have an impact on the happiness of adolescents. Uusitalo-Malmivaara (2014) [74] found that more leisure time increases happiness levels. On the other hand, some studies have shown that an increase in leisure time increases the frequency of gambling and lowers the quality of life [24,75]. Manolis and Roberts (2012) [22] emphasized a moderate amount of leisure time by finding that too much or too little leisure time has a negative impact on well-being.

Along with leisure time, leisure costs are a major predictor of happiness. In general, consumption expenditure affects happiness [76,77]. Among the various consumer expenditures such as food, housing, and healthcare, leisure consumption expenditure is more closely related to happiness [78,79]. Sakkthivel (2011) [80] found that people who engage in non-monetary leisure activities are happier than those who engage in monetary leisure activities. Bo-Ram Kim and Mae Kim (2020) [20] found that there was no significant relationship between leisure consumption expenditure and happiness among adolescents.

Leisure space and location are also important factors. Participation in outdoor activities rather than in indoor activities increases happiness levels [21,81,82], especially outdoor activities that are in nature, contribute significantly to raising the level of happiness in adolescents [83,84]. Benita and Bansal, Tunçer (2019) [85] found that students visiting parks and community centers were more likely to experience subjective well-being compared to students visiting commercial areas. Public leisure facilities, in particular, are closely related to the physical activity of young people. Lee, Kuo, and Chan (2016) [86] found that difficulty accessing public facilities reduced physical activity in adolescents. Ries, Yan, and Voorhees (2011) [23] also emphasized the importance of public utilities in adolescents' physical activity.

On the other hand, adolescents experience a conflict between study and leisure. These conflicts reduce adolescents' academic performance, increase levels of depression, and even lower their quality of life [29]. Therefore, balancing and harmonizing between study and leisure is very important for young people. According to a study that observed the level of happiness between study and leisure, adolescents who balance study and leisure without being biased towards either study or leisure are happier [20,87].

Perceptions and attitudes toward leisure can have a profound effect on a youth's happiness. Lepp (2018) [28] found that adolescents with a high perception of the value of leisure had higher levels of happiness. Positive attitudes towards leisure also increase leisure satisfaction [88–90], which can be found to increase happiness levels [26,27].

The leisure factors that affect the happiness of adolescents discussed so far are summarized in Table 1.

Based on the trends of the studies presented in the previous sections, this study set the following research questions.

**Table 1. Leisure factors affecting the happiness of adolescents.**

| Factor | Previous studies |
|---|---|
| Diversity of leisure activities | Dahan-Oliel et al(2012) [52] <br> Santini et al(2020) [17] |
| Type of leisure activities | Ito et al(2019) [56] <br> Lee et al(2017) [14] <br> Shin & You(2013) [18] |
| Presence or absence of leisure companions | Gray(2011) [68] <br> Lam & McHale(2015) [69] <br> Larson & Csikszentmihalyi(1978) [70] <br> Parker et al(2022) [16] |
| frequency of leisure activities | Doerksen et al(2014) [73] <br> Moljord et al(2011) [15] <br> Te Velde et al(2018) [71] <br> Toker & Kalıpçı(2021) [72] |
| Amount of leisure time | Manolis &Roberts(2012) [22] <br> Moore & Ohtsuka(2000) [75] <br> Uusitalo-Malmivaara(2014) [74] <br> Wang et al(2011) [24] |
| Amount of leisure cost | Kim & Kim(2020) [20] |
| Leisure space | Bailey & Fernando(2012) [21] <br> Leonard(2015) [81] <br> Weng & Chiang(2014) [82] <br> Benita et al(2019) [85] |
| Balance between study and leisure | Lee & Kim(2021) [87] <br> Ratelle et al(2005) [29] |
| Degree of leisure perception and attitude | Bailey et al(2016) [26] <br> Gökyürek(2016) [27] <br> Lepp(2018) [28] |

First, what are the relationships between various leisure factors and the happiness of adolescents?

Second, what are the main leisure factors that affect the happiness of adolescents?

## Research method

The objective of this study is to explore major leisure factors that affect the happiness of adolescents. To achieve this research goal, the following procedures and methods were employed and performed (see Fig 1).

### Research subjects

This study used data from the National Leisure Activity Survey 2019. The National Leisure Activity Survey is a national statistic produced by the Ministry of Culture, Sports and Tourism in the Republic of Korea. The purpose of this survey is to analyze people's demand for leisure activities and the actual situation to identify changes in lifestyle and quality of life level, and to use them as basic data for the formulation of related policies [91,92]. The population of the 2019 National Leisure Activity Survey is 10,060 men and women aged 15 and older in 17 cities and provinces nationwide.

### Research structure

- Investigation agency: Ministry of Culture, Sports and Tourism, Republic of Korea

- Survey target: Population aged 15 and over nationwide

- Effective number of respondents: 10,060

- Survey period: September 9, 2019 ~November 14, 2019

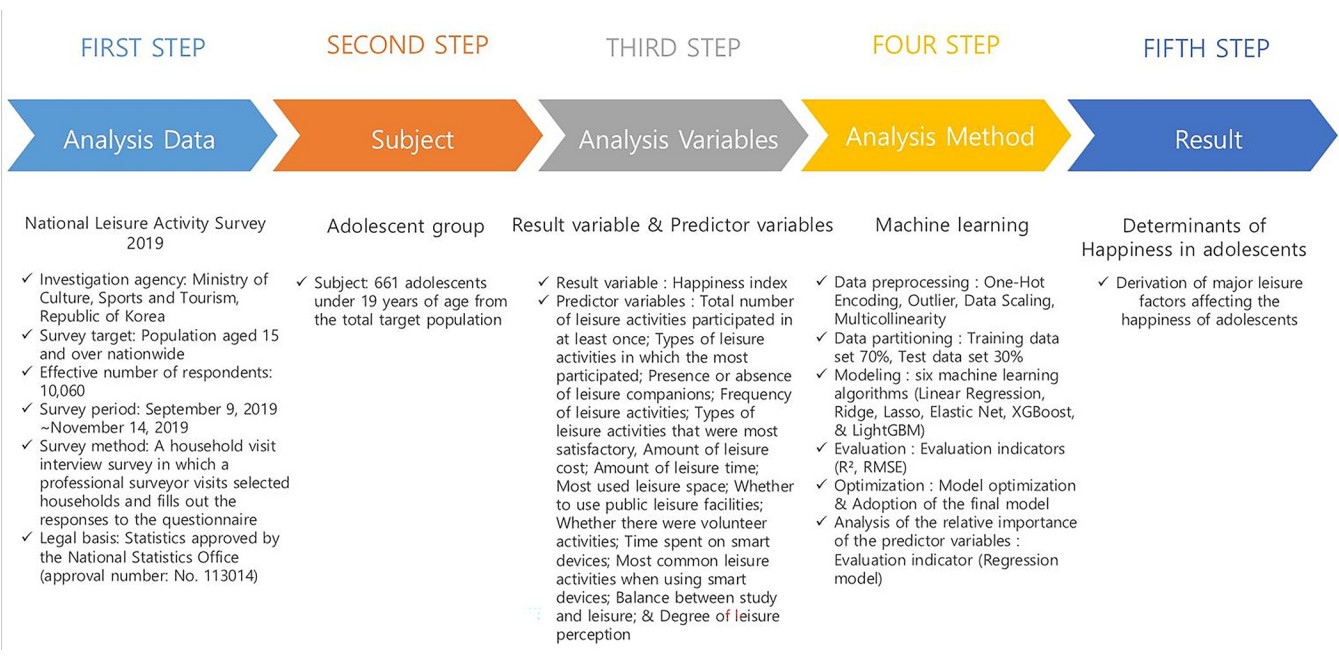

**Fig 1. Research procedures and methods.**

- Survey method: A household visit interview survey in which a professional surveyor visits selected households and fills out the responses to the questionnaire

- Legal basis: Statistics approved by the National Statistics Office (approval number: No. 113014)

## Sampling design

- Target population: Population aged 15 and over nationwide

- Survey population: Household members aged 15 years or older residing in all households in Korea at the time of the survey

- Sampling frame: Based on data from the 2017 Census survey by Statistics Korea

- Stratification: A total of 17 cities and provinces are stratified into large cities and rural areas which divided into Eup and Myeon (i.e., small towns & villages) reflecting the characteristics of urban and rural areas.

- Sampling method: Random extraction after allocating the number of households in each province by square root proportional distribution considering the precision and appropriateness of the sample based on Stratified Multi-Stage Cluster Sampling

**Weight calculation.** Final weight calculation: Design weight*Nonresponse adjustment coefficient*(1/In-household extraction rate)*Population information adjusted coefficient

**Final research subjects for the study.** This study included 661 adolescents under 19 years of age from the total population. The general characteristics of the study subject are shown in Table 2.

According to the 2023 youth statistics of The Ministry of Gender Equality and Family of the Republic of Korea [93], education hour and the leisure time of adolescents in Korea are as follows. With regard to the general characteristics of the research group for the study (i.e., adolescents in Korea), the participation rate of elementary, middle, and high school students in private education in 2022 was 78.3%, and the average private education participation time per

**Table 2. Demographic characteristics of the study subjects.**

| Characteristics | Division | n | % |
|---|---|---|---|
| Gender | Male | 355 | 53.7 |
| | Female | 306 | 46.3 |
| Age | 15 years old | 151 | 22.8 |
| | 16 years old | 128 | 19.4 |
| | 17 years old | 143 | 21.6 |
| | 18 years old | 133 | 20.1 |
| | 19 years old | 106 | 16.0 |
| Number of household members living together | 2 or less | 47 | 7.1 |
| | 3 | 182 | 27.5 |
| | 4 | 337 | 51.0 |
| | 5 or more | 95 | 14.4 |
| Region | Large cities | 285 | 43.1 |
| | Medium and small cities | 285 | 43.1 |
| | Small towns and villages (i.e., Eup, Myeon areas) | 91 | 13.8 |

week was 7.2 hours. In 2022, 23.1% of students in elementary (4th~6th), middle and high school students answered that they study for 2~3 hours a day, excluding regular class hours on weekdays, followed by 1~2 hours (19.4%), 3~4 hours (18.1%) followed by less than 1 hour (17.2%). In the case of leisure time, in 2022, the free time that elementary (4th~6th), middle and high school students can freely use on weekdays is 2~3 hours (24.1%), 1~2 hours (23.5%), and 3~4 hours (17.2%) followed by 5 hours or more (13.8%). In 2022, the average weekly internet usage time for teenagers was 24.3 hours, and for those in their 20s, it was 33.4 hours. In 2022, the main purposes of Internet use by teenagers are leisure activities, communication, data and information acquisition, education and learning, and for those in their 20s, they were leisure activities, communication, data and information acquisition, and website operation [93].

**Selection of variables and result variables.**   In this study, some variables from the National Leisure Activity Survey were extracted and used for analysis. Variable selection was based on prior studies that observed the relationship between leisure and happiness in adolescents.

As a result variable, the 'happiness index' was selected. The happiness index is, 'How happy do you think you are right now?' is a survey about. The questionnaire is structured to respond on a 10-point scale (1 = 'very unhappy', 10 = 'very happy'), and higher response values indicate a higher level of happiness.

**Predictor variables.**   The predictor variables were: the total number of leisure activities participated in at least once, the type of leisure activities in which the most participated, the presence or absence of leisure companions, the frequency of leisure activities, the type of leisure activities that were most satisfactory, the amount of leisure cost, the amount of leisure time, the most used leisure space, whether to use public leisure facilities, whether there were volunteer activities, the time spent on smart devices, the most common leisure activities when using smart devices, the balance between study and leisure, and the degree of leisure perception'. Some of the selected variables were used for analysis as they were, and some were processed by reclassifying the variable values into higher categories or processing them by averaging and totaling in consideration of the imbalance in response frequency and convenience of interpreting the results. Specific information about the predictors is shown in Table 3.

**Research model.**   In this study, the study model as shown in Fig 2 was constructed through a variable selection process.

## Analysis method

This study applied machine learning to explore the main leisure factors that affect adolescents' happiness. Machine learning is considered the core of data science [94] and produces accurate predictions [95]. Machine learning has the advantage of requiring relatively little intervention from researchers and enabling more accurate predictions and decisions compared to traditional statistical techniques in that it learns rules and patterns from existing data to predict the outcome of new data [96]. The specific analysis method is shown in Fig 3.

**Data pre-processing.**   Data preprocessing is an important area of machine learning techniques. Depending on what data you enter, the results can vary greatly. This study treated missing data, dummy variables, outliers, data scaling, and multicollinearity. The preprocessing process of data is described in detail below.

**On-hot encoding.**   One-Hot Encoding on categorical data was performed. Then, considering the multicollinearity problem between the transformed variables, one dummy variable was removed. The list of removed variables (reference variables) is shown in Table 4.

**Table 3. Information about predictor variables.**

| Variable type | Description |
|---|---|
| continuous variable | Total number of leisure activities participated in at least once in the past year(n) |
| categorical variable | Type of leisure activities in which the most participated (Active leisure: 1, Passive leisure: 2) |
| categorical variable | Presence or absence of leisure companions (Without a companion: 1, With a companion: 2) |
| categorical variable | Frequency of leisure activities (Everyday: 1, Not every day: 2) |
| categorical variable | Type of leisure activities that were most satisfactory (Cultural and artistic activity: 1, Sports activity: 2, Tourism: 3, Hobbies: 4, Rest activity: 5, Social activity: 6) |
| continuous variable | Monthly average leisure cost(Won) |
| continuous variable | Daily average leisure time(Hour) |
| categorical variable | Most used leisure space outside of the home (Indoor space: 1, Outdoor space: 2) |
| categorical variable | Whether to use public leisure facilities (Yes: 1, No: 2) |
| categorical variable | Whether there were volunteer activities (Yes: 1, No: 2) |
| continuous variable | Time spent on smart devices (Hour) |
| categorical variable | Most common leisure activities when using smart devices (Game:1, SNS: 2, Others: 3) |
| continuous variable | Balance between study and leisure (On a 7 point Likert scale) |
| continuous variable | Degree of leisure perception (On a 7 point Likert scale) |

**Outliers.** Values that are abnormally apart from other values, that is, values that are too small or too large, can significantly affect the data analysis results, so outliers are a problem that must be solved in the process of data analysis. The criteria for judging outliers were the interquartile range (IQR = Inter-Quartile Range), and values less than quartile-IQR * 1.5 and values greater than the third quartile + IQR * 1.5 were removed. As a result, 40 pieces of data were removed.

**Data scaling.** The data was converted to adjust the units of different data equally. Data scaling applied the standardization method to convert the values of all data to a standard normal distribution with a mean of 0 and a variance of 1.

**Multicollinearity.** To identify the multicollinearity problem, VIF (Variance Inflation Factor) was identified. The multicollinearity criterion was VIF 10 or higher. As a result of calculating VIF, the maximum VIF was 3.97, confirming that there was no multicollinearity problem between the predictors.

**Data partitioning.** In order to generalize the predictive model, the entire data was divided into train data set and evaluation data (i.e., test data set). The data for training is used when training the model and the data for evaluation is data that evaluates the performance of the trained model. The model built from the training data was applied to the evaluation data to evaluate the generalizability of the model. The data division ratio was applied as 70% for training data and 30% for evaluation data.

**Model and evaluation.** A prediction model was built based on the supervised learning algorithm of machine learning, and the performance of the model was evaluated. The process

**Predictor variables**

- Total number of leisure activities participated in at least once
- Type of leisure activities in which the most participated
- Presence or absence of leisure companions, the frequency of leisure activities
- Type of leisure activities that were most satisfactory
- Amount of leisure cost
- Amount of leisure time
- Most used leisure space
- Whether to use public leisure facilities
- Whether there were volunteer activities
- Time spent on smart devices
- Most common leisure activities when using smart devices
- Balance between study and leisure
- Degree of leisure perception

**Result variable**

Happiness index

**Fig 2. Research model.**

of creating a regression prediction model is a process of optimizing the coefficients of the independent variables by minimizing the difference between the predicted value and the actual value, that is, the value of the cost function. In this study, general linear regression, the most representative regression model, and regulated linear models (i.e., Ridge Regression, Lasso Regression, Elastic Net Regression), which are improvement models of general linear regression, were used. The regularized regression model is a method of increasing the weight of variables of high importance and lowering the weight of variables of low importance to increase the possibility of generalization by preventing multicollinearity between variables and overfitting of the prediction model. In addition, the ensemble model of eXtreme Gradient Boost (XGB) and Light Gradient Boosting Machine (LGBM) was used. An ensemble model is a method of predicting results by merging different prediction models. As such, a total of six machine learning algorithms (Linear Regression, Ridge Regression, Lasso Regression, Elastic Net Regression, eXtreme Gradient Boost, & Light Gradient Boosting Machine) were used in this study.

The evaluation indicators used were $R^2$ (r-squared) and RMSE (root mean square error), which are representative performance indicators of regression models. $R^2$ refers to a coefficient that measures how much the regression line estimated by the regression analysis explains the actual sample. If this value is 1, it means that the regression line is in perfect agreement with the data. Conversely, if the coefficient of determination is 0, it means that the regression line does not explain the distribution of the data at all. RMSE is a generalized measure of standard

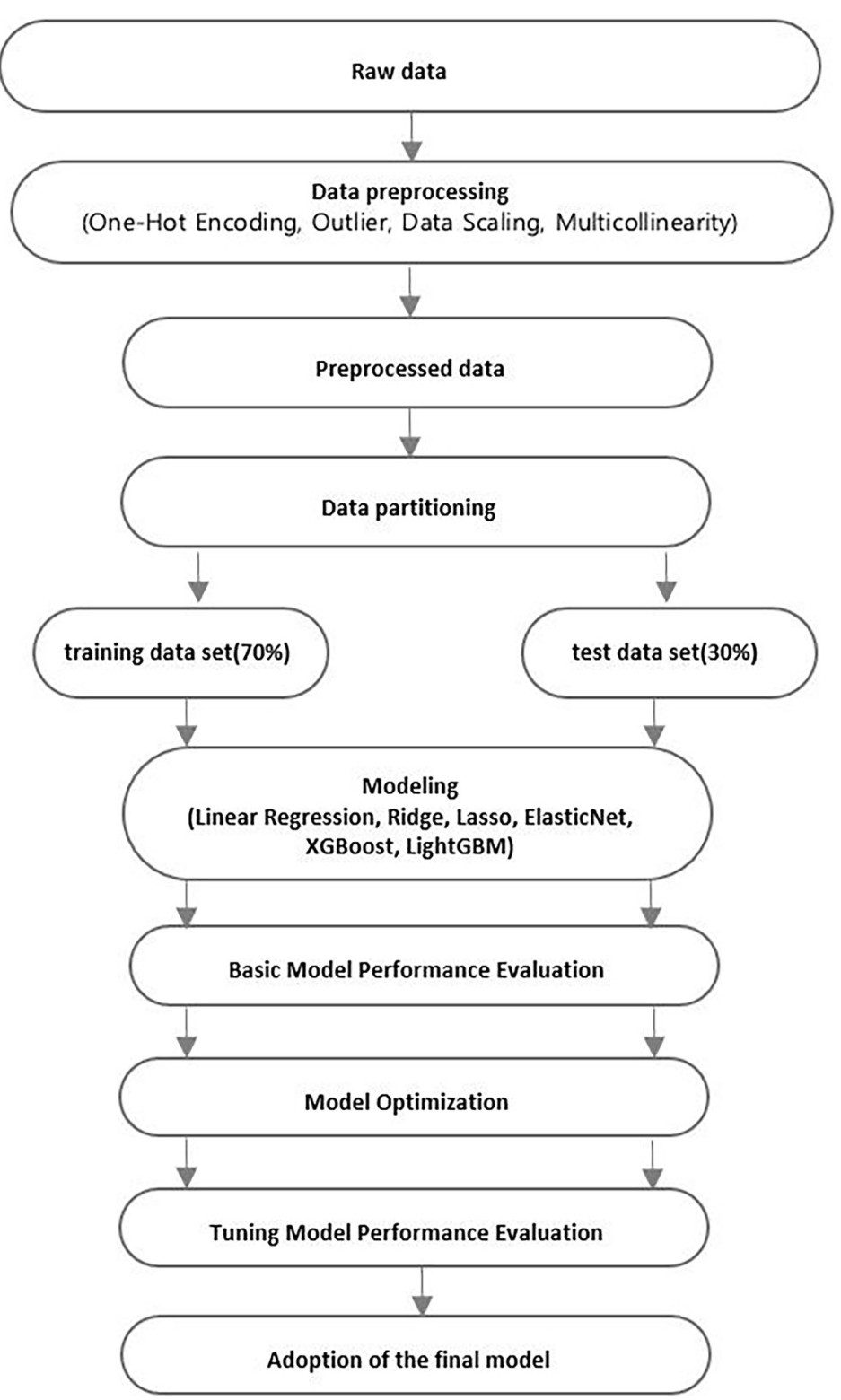

**Fig 3. Analysis method.**

**Table 4. List of baseline variables.**

| Name of variables | Baseline variables |
|---|---|
| type of leisure activities in which the most participated | Passive leisure |
| Ways to participate in leisure activities | Alone |
| frequency of leisure activities | Not everyday |
| type of leisure activities that were most satisfactory | Rest activity |
| Most used leisure space outside of the home | Indoor space |
| whether to use public leisure facilities | No |
| whether there were volunteer activities | No |
| Most common leisure activities when using smart devices | Others |

deviation and represents the average difference between the value predicted by the prediction model and the actual value. Therefore, the lower the value of root mean square error, the better the prediction model.

## Model optimization

Hyper parameters were optimized to improve the performance of previously built predictive models. As a method for optimizing hyper parameters, grid search was used. Grid search refers to a method of specifying certain values for each hyper parameter and learning data for all combinations of specified values to search for a combination of hyper parameters that represent optimal performance indicators. In this study, the hyper parameters were optimized for each algorithm, and the model with the best performance was adopted as the final model.

## Evaluation of the importance of predictor variables

The relative importance of the predictor variables was evaluated based on the final model. The evaluation indicator used was a regression coefficient that indicates the influence of the predictor variable. Among the predictive factors entered into the model, three factors with high regression coefficients were extracted to identify the determinants of happiness.

## Analysis tools

The analysis tool utilized Python 3.7 and the jupyter notebook.

# Research results

## Model performance evaluation

The performance of the predictive model using six machine learning algorithms (LinearRegression, Ridge, Lasso, ElasticNet, XGBoost, and LightGBM) is shown in Table 5 and Fig 4.

**Table 5. Evaluating model performance.**

| Model | $R^2$ | RMSE |
|---|---|---|
| LinearRegression | 0.1150 | 1.3194 |
| Ridge | 0.1153 | 1.3191 |
| Lasso | -0.0199 | 1.4164 |
| ElasticNet | -0.0199 | 1.4164 |
| XGBoost | -0.1569 | 1.5085 |
| LightGBM | -0.0982 | 1.4697 |

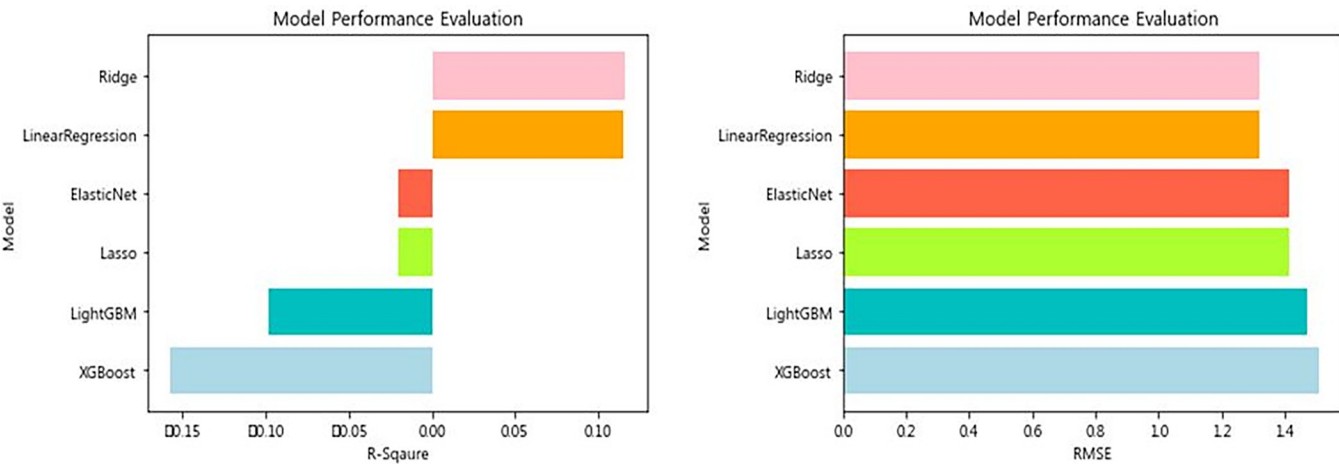

**Fig 4. Model performance evaluation.**

The highest-performing models were the Ridge model ($R^2$ = 0.1153, RMSE = 1.3191), the Linear Regression model ($R^2$ = 0.1150, RMSE = 1.3194), the Elastic Net model ($R^2$ = -0.0199, RMSE = 1.4164), the Lasso model ($R^2$ = -0.0199, RMSE = 1.4164), the LightGBM model ($R^2$ = -0.0982, RMSE = 1.5085), and the XGBoost model ($R^2$ = -0.1569, RMSE = -0.1569).

## Model performance optimization

Hyper parameters were tuned to optimize the performance of the machine learning model. After hyper parameter tuning, the model's performance is shown in Table 6 and Fig 5. The highest-performing models were ($R^2$ = 0.121, RMSE = 1.315), Lasso ($R^2$ = 0.119, RMSE = 1.316), Linear Regression ($R^2$ = 0.115, RMSE = 1.319), Elastic Net model ($R^2$ = 0.108, RMSE = 1.325), XGBoost model ($R^2$ = 0.102, RMSE = 1.329), and LightGBM model ($R^2$ = 0.079, RMSE = 1.346). The performance of these models is somewhat low compared to previous studies that used machine learning to predict happiness [97–99].

## Evaluation of the importance of predictors

The importance of predictors was assessed to explore the main factors influencing adolescent happiness. Fig 6 extracts three most important predictors from the model. Based on the Ridge

**Table 6. Model performance optimization.**

| Model | Hyperparameter | $R^2$ | RMSE |
|---|---|---|---|
| LinearRegression | - | 0.115 | 1.319 |
| Ridge | alpha = 45 | 0.121 | 1.315 |
| Lasso | alpha = 0.01 | 0.119 | 1.316 |
| ElasticNet | alpha = 0.1 | 0.108 | 1.325 |
| XGBoost | n_estimators = 500<br>learning_rate = 0.01<br>colsample_bytree = 0.8<br>subsample = 0.5<br>max_depth = 1<br>min_child_weight = 2<br>colsample_bylevel = 1<br>gamma = 1<br>reg_lambda = 0.01 | 0.102 | 1.329 |
| LightGBM | learning_rate = 0.01, n_estimators = 500, max_depth = 1 | 0.079 | 1.346 |

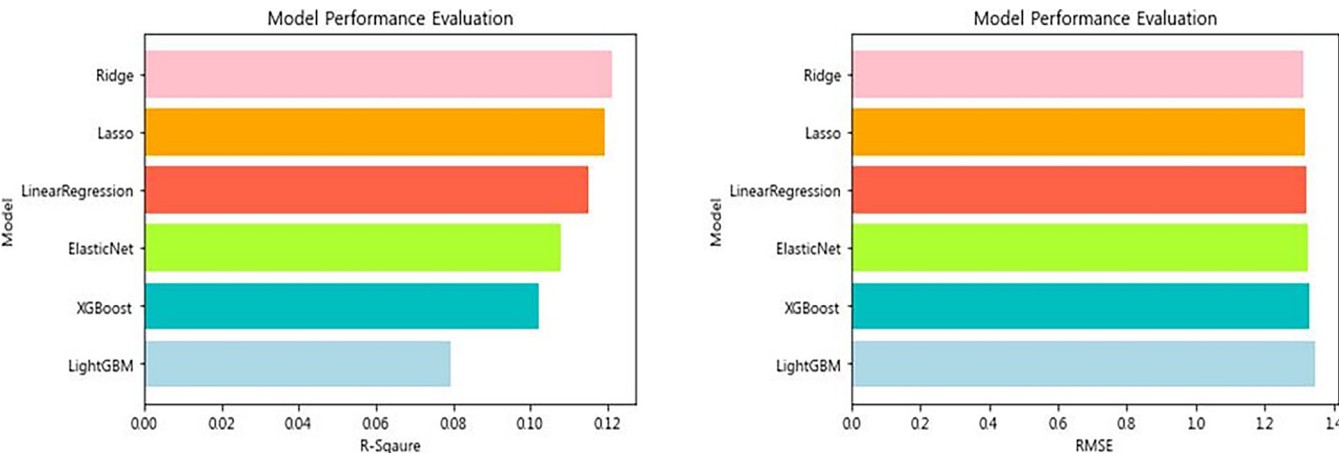

**Fig 5. Model performance evaluation.**

**Fig 6. Evaluation of the importance of predictors.**

model, which showed the highest performance, the most important predictors were leisure perception, balance between study and leisure, and public leisure space.

## Discussion

Many studies have shown that various leisure factors, such as the type of leisure, time, cost, and space, affect the happiness of adolescents. Little is known, however, about which of these factors is more of the key factor in youth's happiness. Therefore, the leisure factors that determine the happiness of adolescents were explored by examining the relationship between various leisure factors and happiness in this research.

The most important finding was that leisure perception had the greatest influence on adolescents' happiness. A rather surprising result, considering that the lack of leisure time has been pointed out as the biggest problem in leisure activities of adolescents [91,92,100]. This discovery is significant in that it shows how important it is in the lives of adolescents to understand the nature of leisure. Furthermore, these results remind us of the importance of youth leisure education. Leisure education plays an important role in raising adolescents' leisure awareness [100–104]. Therefore, schooling should instill in young people a correct perception of leisure through leisure education. So far, many studies have been conducted on leisure in adolescents, but relatively few studies have highlighted the importance of leisure perception. Leisure awareness is a topic that should be seriously studied in adolescent leisure research. In the future, more research focusing on leisure perception is needed.

In this study, the balance between study and leisure was found to be the main factor affecting the happiness of adolescents. Interestingly, the more we focused on leisure, the higher our level of happiness. These results are consistent with those of Kim and Kim (2020) [20], who found that adolescents who are more immersed in leisure between study and leisure have higher levels of happiness than adolescents who are more immersed in schoolwork. This also supports research showing that excessive study time and tutoring negatively affect the lives of adolescents [105,106]. In this study, it has been found that between study and leisure, leisure may be a more important factor in a youth's happiness. This finding can have important implications for adolescents who are experiencing conflicts between academics and leisure.

Another important finding was that leisure space is a major factor influencing the happiness of adolescents. Adolescents who use public leisure spaces have been found to have higher levels of happiness. A possible explanation for these results may be that public leisure spaces serve an important function as places for adolescents' physical and social activities [23,85,107,108]. Consistent to these findings, Benita, Bansal, and Tunçer (2019) [85] found that students who visit parks and community centers are more likely to experience subjective well-being than students who visit more commercial areas. Benita and Bansal, Tunçer (2019) [85] and the results of this research suggest that public leisure spaces are a key factor in improving the level of happiness in adolescents. However, since the concepts of public and private space are unclear and their boundaries are fluid [109], extreme caution should be exercised when interpreting these results. With the privatization and commercialization of public space, leisure space for young people is gradually disappearing [109–111]. The findings of this study will help us better understand the importance of public space.

In this regard, the findings of this study imply that it would be desirable to expand the current results in the future with an in-depth analysis of adolescents' characteristics, such as gender and school level, and to conduct longitudinal studies that can help understand how leisure factors that affect adolescents' happiness behave over time.

## Conclusion

The purpose of this study is to establish a happiness prediction model for adolescents based on machine learning, and to identify the relative importance of predictive factors to derive major leisure factors that affect the happiness of adolescents. The conclusions obtained through this study are as follows.

This study found that leisure perceptions, academic and leisure balance, and public leisure spaces all have a very important impact on adolescents' well-being. These findings will help other researchers investigating the relationship between leisure and happiness in adolescents design their studies. Furthermore, these results will have a positive impact on youth leisure policymakers. Many countries recognize the importance of leisure activities for youth and are making related policy efforts. In this way, the findings of this research imply that rather than leaving leisure activities to individual youth, there is a need to support them institutionally at the national level. In particular, as the importance of public leisure spaces has been highlighted in this study, there is a need to expand public leisure facilities for youth.

However, this study has some limitations from an academic standpoint. Firstly, this study used quantitative research methods. Quantitative research methods have limitations in providing an in-depth understanding of the experiences, attitudes, motivations, and contexts of the research subjects. In particular, most of the data used in this study are multiple-choice surveys, so there is a limitation in not being able to specifically express the opinions or thoughts of the subjects. Considering this, qualitative research methods also need to be considered for future studies. Secondly, since this study used domestic data to fully investigate the domestic situation on the relationship between leisure factors and adolescents' happiness within the Korean context, it is difficult to generalize the results of the study internationally. Although the results and claims made in this study are supported by foreign studies which were conducted internationally and also by explanations about the domestic cultural context, it would be desirable to extend the current results with future international data analyses in future studies so that more meaningful and comprehensive findings can be derived.

In addition, longitudinal studies to understand how the leisure factors that affect adolescent happiness change over time would be required. Lastly, this study has limitations in in-depth discussion of the relationship between happiness and leisure among adolescents. In this regard, in the future, it would be necessary to expand the current results through in-depth analysis according to the characteristics of adolescents, such as gender, school level, and region of residence.

Despite these limitations, it is significant that this study explored the leisure factors that determine happiness in adolescents using reliable and nationally representative national statistics. In particular, it is significant that by using machine learning, which is considered to be the core of predictive analytics, it was possible to improve the accuracy of research results. Finally, the findings of this research will serve as a basis for formulating and promoting youth leisure policies.

## Supporting information

**S1 File. National leisure activity survey 2019 raw data.**
(XLSX)

**S2 File. National leisure activity survey codebook 2019.**
(XLSX)

## Author Contributions

**Conceptualization:** Eui-Jae Kim, Hyun-Wook Kang.

**Data curation:** Eui-Jae Kim.

**Formal analysis:** Eui-Jae Kim.

**Investigation:** Eui-Jae Kim.

**Methodology:** Eui-Jae Kim.

**Project administration:** Hyun-Wook Kang.

**Resources:** Eui-Jae Kim.

**Software:** Eui-Jae Kim.

**Supervision:** Hyun-Wook Kang.

**Validation:** Eui-Jae Kim, Hyun-Wook Kang.

**Visualization:** Eui-Jae Kim.

**Writing – original draft:** Eui-Jae Kim, Hyun-Wook Kang.

**Writing – review & editing:** Seong-Man Park.

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
