## [Decision Letter · Decision Letter 0]

19 Dec 2023

PONE-D-23-32346Determinants of the Happiness of Adolescents: A Leisure PerspectivePLOS ONE

Dear Dr. Park,

Thank you for submitting your manuscript to PLOS ONE. After careful consideration, we feel that it has merit but does not fully meet PLOS ONE’s publication criteria as it currently stands. Therefore, we invite you to submit a revised version of the manuscript that addresses the points raised during the review process.

We look forward to receiving your revised manuscript.

Kind regards,

Henri Tilga, PhD

Academic Editor

PLOS ONE

5. We notice that your supplementary figures are uploaded with the file type 'Figure'. Please amend the file type to 'Supporting Information'. Please ensure that each Supporting Information file has a legend listed in the manuscript after the references list.

Additional Editor Comments:

The Reviewer has provided several useful comments to increase the quality of this manuscript. Please carefully follow all the comments made by the Reviewer and revise the manuscript accordingly.

Reviewers' comments:

Reviewer's Responses to Questions

**Comments to the Author**

1. Is the manuscript technically sound, and do the data support the conclusions?

Reviewer #1: Partly

2. Has the statistical analysis been performed appropriately and rigorously? 

Reviewer #1: No

3. Have the authors made all data underlying the findings in their manuscript fully available?

Reviewer #1: Yes

4. Is the manuscript presented in an intelligible fashion and written in standard English?

Reviewer #1: Yes

5. Review Comments to the Author

Reviewer #1: The article "Determinants of the Happiness of Adolescents: A Leisure Perspective" addresses an important and timely topic—the impact of leisure factors on the happiness of adolescents. The use of machine learning to analyze the National Leisure Activity Survey 2019 data adds a contemporary and sophisticated analytical approach to the study. However, there are both strengths and weaknesses that warrant critical consideration.

Strengths:

The study's objective, as outlined, is clear and relevant, aiming to explore the relationship between various leisure factors and adolescent happiness. This aligns with the growing interest in understanding the well-being of youth, making the study socially and academically relevant. The article provides a detailed overview of the methodology, including the use of machine learning and the specific dataset employed (National Leisure Activity Survey 2019). This transparency enhances the credibility and replicability of the study. The findings indicating the importance of leisure perceptions, academic and leisure balance, and public leisure space on adolescents' well-being contribute valuable insights. This identification of key factors can guide policymakers, educators, and parents in promoting positive leisure experiences for adolescents.

While it mentions the use of machine learning, it neds to include more specific details about the machine learning methods employed. Providing information about the algorithms, features, and model validation would enhance the transparency of the analysis and help readers assess the robustness of the findings.

On the other hand, the reliance on survey data, as outlined in the methodology, introduces potential limitations such as self-reporting bias and the inability to capture nuanced aspects of leisure experiences. Authors should emphasize these limitations and discuss their implications for the generalizability of the findings.

The study focuses on data from the Republic of Korea, limiting the generalizability of its findings to a global context. The article should explicitly acknowledge this geographical specificity and discuss how cultural and regional differences may impact the results. While findings may contribute to a better understanding of leisure and happiness in adolescents, they need to influence and be specific regarding practical implications for policy development or interventions. Providing concrete recommendations based on the study's findings would enhance its practical relevance.

1. Enhance methodological detail and provide more detailed information about the specific machine learning techniques employed, including the choice of algorithms, feature selection, and model evaluation metrics.

2. Address survey limitations by acknowledging the inherent limitations of survey-based research, including potential biases, and discuss how these limitations may impact the study's findings and their generalizability.

3. Consider cultural context while explicitly recognizing the cultural and geographical specificity of the study and discuss how these factors may influence the observed relationships between leisure factors and adolescent happiness.

4. Provide practical recommendations while clearly articulating them based on the study's findings, particularly in terms of policy implications and interventions that could improve adolescents' leisure experiences and well-being.

5. Include a table of authors at the end of the literature review section to analyze the most relevant paper in each variable

6. Include a diagram to illustrate the applied methodological process to your research

7. Expand your conclusions including your future work and the relevance in the managerial, academic, and government arena.

8. The authors need to explain further each of the six machine learning algorithms (LinearRegression, Ridge, Lasso, ElasticNet, XGBoost, and LightGBM)

In conclusion, while the article contributes valuable insights into the relationship between leisure factors and adolescent happiness, addressing the highlighted weaknesses and implementing the recommended improvements would strengthen the overall quality and impact of the research.

6. PLOS authors have the option to publish the peer review history of their article (what does this mean?). If published, this will include your full peer review and any attached files.

Reviewer #1: No

---

## [Author Response · Author response to Decision Letter 0]

30 Jan 2024

Dear. Dr. Henri Tilga, Academic Editor

First of all, we really appreciate your kind and strong support on the manuscript submission.

As per your and reviewers’ revision request, we revised our manuscript so that the revised version of the manuscript fully addresses the points raised during the review process.

Here are the responses and revisions. 

Title of the manuscript

Determinants of the Happiness of Adolescents: A Leisure Perspective

Data availability statement

This study used the method of machine learning to analyze national statistical data, National Leisure Activity Survey, which is open to the public. The data used in this study were from the National Leisure Activity Survey 2019, which is a national statistic produced by the Ministry of Culture, Sports and Tourism in the Republic of Korea (Statistics approved by the National Statistics Office - approval number: No. 113014).

Reviewer comments

1. Enhance methodological detail and provide more detailed information about the specific machine learning techniques employed, including the choice of algorithms, feature selection, and model evaluation metrics.

Response: We added the following sections in the method section.

Model and evaluation 

A prediction model was built based on the supervised learning algorithm of machine learning, and the performance of the model was evaluated. The process of creating a regression prediction model is a process of optimizing the coefficients of the independent variables by minimizing the difference between the predicted value and the actual value, that is, the value of the cost function. In this study, general linear regression, the most representative regression model, and regulated linear models (i.e., Ridge Regression, Lasso Regression, Elastic Net Regression), which are improvement models of general linear regression, were used. The regularized regression model is a method of increasing the weight of variables of high importance and lowering the weight of variables of low importance to increase the possibility of generalization by preventing multicollinearity between variables and overfitting of the prediction model. In addition, the ensemble model of eXtreme Gradient Boost (XGB) and Light Gradient Boosting Machine (LGBM) was used. An ensemble model is a method of predicting results by merging different prediction models. As such, a total of six machine learning algorithms (Linear Regression, Ridge Regression, Lasso Regression, Elastic Net Regression, eXtreme Gradient Boost, & Light Gradient Boosting Machine) were used in this study. 

The evaluation indicators used were R² (r-squared) and RMSE (root mean square error), which are representative performance indicators of regression models. R² refers to a coefficient that measures how much the regression line estimated by the regression analysis explains the actual sample. If this value is 1, it means that the regression line is in perfect agreement with the data. Conversely, if the coefficient of determination is 0, it means that the regression line does not explain the distribution of the data at all. RMSE is a generalized measure of standard deviation and represents the average difference between the value predicted by the prediction model and the actual value. Therefore, the lower the value of root mean square error, the better the prediction model. 

Evaluation of the importance of predictor variables

The relative importance of the predictor variables was evaluated based on the final model. The evaluation indicator used was a regression coefficient that indicates the influence of the predictor variable. Among the predictive factors entered into the model, three factors with high regression coefficients were extracted to identify the determinants of happiness.

2. Address survey limitations by acknowledging the inherent limitations of survey-based research, including potential biases, and discuss how these limitations may impact the study's findings and their generalizability.

Response: We added the limitations at the conclusion section as follows:

However, this study has some limitations from an academic standpoint. Firstly, this study used quantitative research methods. Quantitative research methods have limitations in providing an in-depth understanding of the experiences, attitudes, motivations, and contexts of the research subjects. In particular, most of the data used in this study are multiple-choice surveys, so there is a limitation in not being able to specifically express the opinions or thoughts of the subjects. Considering this, qualitative research methods also need to be considered for future studies.

3. Consider cultural context while explicitly recognizing the cultural and geographical specificity of the study and discuss how these factors may influence the observed relationships between leisure factors and adolescent happiness.

Response: We added the limitations at the conclusion section as follows:

Secondly, since this study used domestic data to fully investigate the domestic situation on the relationship between leisure factors and adolescents’ happened within the Korean context, it is difficult to generalize the results of the study internationally. Although the results and claims made in this study are supported by foreign studies which were conducted internationally and also by explanations about the domestic cultural context, it would be desirable to extend the current results with future international data analyses in future studies so that more meaningful and comprehensive findings can be derived. 

4. Provide practical recommendations while clearly articulating them based on the study's findings, particularly in terms of policy implications and interventions that could improve adolescents' leisure experiences and well-being.

Response: We added the following at the conclusion section as follows:

Many countries recognize the importance of leisure activities for youth and are making related policy efforts. In this way, the findings of this research imply that rather than leaving leisure activities to individual youth, there is a need to support them institutionally at the national level. In particular, as the importance of public leisure spaces has been highlighted in this study, there is a need to expand public leisure facilities for youth.

5. Include a table of authors at the end of the literature review section to analyze the most relevant paper in each variable

Response: We added the following table at the end of the literature review section.

Table 1. Leisure factors affecting the happiness of adolescents 

Factor Previous studies

Diversity of leisure activities Dahan-Oliel et al(2012) [54]

Santini et al(2020) [17]

Type of leisure activities Ito et al(2019) [56]

Lee et al(2017) [14]

Shin & You(2013) [18]

Presence or absence of leisure companions Gray(2011) [68]

Lam & McHale(2015) [69] 

Larson & Csikszentmihalyi(2014) [70]

Parker et al(2022) [16]

frequency of leisure activities Doerksen et al(2014) [73]

Moljord et al(2011) [15]

Te Velde et al(2018) [71]

Toker & Kalıpçı(2021) [72]

Amount of leisure time Manolis &Roberts(2012) [22]

Moore & Ohtsuka(2000) [75]

Uusitalo-Malmivaara(2014) [74]

Wang et al(2011) [24]

Amount of leisure cost Kim & Kim(2020) [20]

Leisure space Bailey & Fernando(2012) [21]

Leonard(2015) [81]

Weng & Chiang(2014) [82]

Benita et al(2019) [85]

Balance between study and leisure Lee & Kim(2021) [87]

Ratelle et al(2005) [29]

Degree of leisure perception and attitude

 Bailey et al(2016) [26]

Gökyürek(2016) [27]

Lepp(2018) [28]

6. Include a diagram to illustrate the applied methodological process to your research

Response: We added a diagram to illustrate the applied methodological process at the beginning of the research method section. 

Figure 1. Research procedures and methods

7. Expand your conclusions including your future work and the relevance in the managerial, academic, and government arena.

Response: We expanded our conclusions as follows.

Conclusion 

The purpose of this study is to establish a happiness prediction model for adolescents based on machine learning, and to identify the relative importance of predictive factors to derive major leisure factors that affect the happiness of adolescents. The conclusions obtained through this study are as follows.

This study found that leisure perceptions, academic and leisure balance, and public leisure spaces all have a very important impact on adolescents' well-being. These findings will help other researchers investigating the relationship between leisure and happiness in adolescents design their studies. Furthermore, these results will have a positive impact on youth leisure policymakers. Many countries recognize the importance of leisure activities for youth and are making related policy efforts. In this way, the findings of this research imply that rather than leaving leisure activities to individual youth, there is a need to support them institutionally at the national level. In particular, as the importance of public leisure spaces has been highlighted in this study, there is a need to expand public leisure facilities for youth.

However, this study has some limitations from an academic standpoint. Firstly, this study used quantitative research methods. Quantitative research methods have limitations in providing an in-depth understanding of the experiences, attitudes, motivations, and contexts of the research subjects. In particular, most of the data used in this study are multiple-choice surveys, so there is a limitation in not being able to specifically express the opinions or thoughts of the subjects. Considering this, qualitative research methods also need to be considered for future studies. Secondly, since this study used domestic data to fully investigate the domestic situation on the relationship between leisure factors and adolescents’ happened within the Korean context, it is difficult to generalize the results of the study internationally. Although the results and claims made in this study are supported by foreign studies which were conducted internationally and also by explanations about the domestic cultural context, it would be desirable to extend the current results with future international data analyses in future studies so that more meaningful and comprehensive findings can be derived. 

 In addition, longitudinal studies to understand how the leisure factors that affect adolescent happiness change over time would be required. Lastly, this study has limitations in in-depth discussion of the relationship between happiness and leisure among adolescents. In this regard, in the future, it would be necessary to expand the current results through in-depth analysis according to the characteristics of adolescents, such as gender, school level, and region of residence.

Despite these limitations, it is significant that this study explored the leisure factors that determine happiness in adolescents using reliable and nationally representative national statistics. In particular, it is significant that by using machine learning, which is considered to be the core of predictive analytics, it was possible to improve the accuracy of research results. Finally, the findings of this research will serve as a basis for formulating and promoting youth leisure policies.

8. The authors need to explain further each of the six machine learning algorithms (LinearRegression, Ridge, Lasso, ElasticNet, XGBoost, and LightGBM)

Response: Response: We added the following sections in the method section.

Model and evaluation 

A prediction model was built based on the supervised learning algorithm of machine learning, and the performance of the model was evaluated. The process of creating a regression prediction model is a process of optimizing the coefficients of the independent variables by minimizing the difference between the predicted value and the actual value, that is, the value of the cost function. In this study, general linear regression, the most representative regression model, and regulated linear models (i.e., Ridge Regression, Lasso Regression, Elastic Net Regression), which are improvement models of general linear regression, were used. The regularized regression model is a method of increasing the weight of variables of high importance and lowering the weight of variables of low importance to increase the possibility of generalization by preventing multicollinearity between variables and overfitting of the prediction model. In addition, the ensemble model of eXtreme Gradient Boost (XGB) and Light Gradient Boosting Machine (LGBM) was used. An ensemble model is a method of predicting results by merging different prediction models. As such, a total of six machine learning algorithms (Linear Regression, Ridge Regression, Lasso Regression, Elastic Net Regression, eXtreme Gradient Boost, & Light Gradient Boosting Machine) were used in this study. 

The evaluation indicators used were R² (r-squared) and RMSE (root mean square error), which are representative performance indicators of regression models. R² refers to a coefficient that measures how much the regression line estimated by the regression analysis explains the actual sample. If this value is 1, it means that the regression line is in perfect agreement with the data. Conversely, if the coefficient of determination is 0, it means that the regression line does not explain the distribution of the data at all. RMSE is a generalized measure of standard deviation and represents the average difference between the value predicted by the prediction model and the actual value. Therefore, the lower the value of root mean square error, the better the prediction model. 

Evaluation of the importance of predictor variables

The relative importance of the predictor variables was evaluated based on the final model. The evaluation indicator used was a regression coefficient that indicates the influence of the predictor variable. Among the predictive factors entered into the model, three factors with high regression coefficients were extracted to identify the determinants of happiness.

9. In conclusion, while the article contributes valuable insights into the relationship between leisure factors and adolescent happiness, addressing the highlighted weaknesses and implementing the recommended improvements would strengthen the overall quality and impact of the research.

Response: We did our best to address the highlighted weaknesses and implement the recommended improvements as stated above so that the overall quality and impact of the research can be improved.

Thank you so much for your kind and detailed comments. 

All the best,

Eui Jae Kim, Hyun Wook Kang, Seong Man Park

---

## [Decision Letter · Decision Letter 1]

29 Feb 2024

PONE-D-23-32346R1Determinants of the Happiness of Adolescents: A Leisure PerspectivePLOS ONE

Dear Dr. Park,

Thank you for submitting your manuscript to PLOS ONE. After careful consideration, we feel that it has merit but does not fully meet PLOS ONE’s publication criteria as it currently stands. Therefore, we invite you to submit a revised version of the manuscript that addresses the points raised during the review process.

We look forward to receiving your revised manuscript.

Kind regards,

Henri Tilga, PhD

Academic Editor

PLOS ONE

Journal Requirements:

Reviewers' comments:

Reviewer's Responses to Questions

**Comments to the Author**

1. If the authors have adequately addressed your comments raised in a previous round of review and you feel that this manuscript is now acceptable for publication, you may indicate that here to bypass the “Comments to the Author” section, enter your conflict of interest statement in the “Confidential to Editor” section, and submit your "Accept" recommendation.

Reviewer #1: All comments have been addressed

2. Is the manuscript technically sound, and do the data support the conclusions?

Reviewer #1: Yes

3. Has the statistical analysis been performed appropriately and rigorously? 

Reviewer #1: Yes

4. Have the authors made all data underlying the findings in their manuscript fully available?

Reviewer #1: Yes

5. Is the manuscript presented in an intelligible fashion and written in standard English?

Reviewer #1: Yes

6. Review Comments to the Author

Reviewer #1: The revised manuscript significantly enhances its methodological clarity, transparency, and academic rigor in response to my first reviewer feedback.

By elaborating on the machine learning techniques used, including general linear regression, Ridge Regression, Lasso Regression, Elastic Net Regression, and ensemble models like eXtreme Gradient Boost and Light Gradient Boosting Machine, the study strengthens its methodological foundation.

Including performance indicators like R² and RMSE further enhances the reproducibility of the research. Acknowledging the limitations and potential biases of survey-based research, the manuscript discusses the impact of cultural and geographical specificity, rooted in the Republic of Korea, on the generalizability of findings.

It underscores the importance of considering local dynamics in understanding the relationship between leisure activities and adolescent happiness, while suggesting the complementary role of qualitative approaches for future research.

Practical recommendations for improving adolescent leisure experiences highlight the study's commitment to translating academic insights into actionable strategies.

Enhanced with a comprehensive literature review and a methodological diagram, the revised manuscript offers a holistic approach to exploring adolescent happiness from a leisure perspective, making significant contributions to the field through detailed methodological explanations and contextually grounded insights.

I recommend the following:

1. Include references in your R² and RMSE obtained values to validate with previous studies.

After the previous arguments and after attending the recommendation I recommend this paper to continue with the editorial process.

7. PLOS authors have the option to publish the peer review history of their article (what does this mean?). If published, this will include your full peer review and any attached files.

Reviewer #1: **Yes: **Eduardo Ahumada-Tello, PhD

---

## [Author Response · Author response to Decision Letter 1]

5 Mar 2024

Dear. Dr. Henri Tilga, Academic Editor

With regard to your revision request, we really appreciate your kind and strong support on the manuscript submission. As per your and reviewers’ revision request, we revised our manuscript so that the revised version of the manuscript fully addresses the points raised during the review process.

Here are the responses and revisions. 

Reviewer comments

1. Include references in your R² and RMSE obtained values to validate with previous studies.

Response: We included the following references in our R² and RMSE obtained values to validate with previous studies.

Page 20: Hyper parameters were tuned to optimize the performance of the machine learning model. After hyper parameter tuning, the model's performance is shown in Table 6 and Figure 5. The highest-performing models were (R² = 0.121, RMSE = 1.315), Lasso (R² = 0.119, RMSE = 1.316), Linear Regression (R² = 0.115, RMSE = 1.319), Elastic Net model (R² = 0.108, RMSE = 1.325), XGBoost model (R² = 0.102, RMSE = 1.329), and LightGBM model (R² = 0.079, RMSE = 1.346). The performance of these models is somewhat low compared to previous studies that used machine learning to predict happiness [97-99].

Added references:

97. Airlangga G. Deciphering Urban Happiness: Analysis of Machine Learning Approaches for Comprehensive Urban Planning. Jurasik (Jurnal Riset Sistem Informasi dan Teknik Informatika). 2024; 9(1): 345-353.

98. Pan Z, Cutumisu M. Using machine learning to predict UK and Japanese secondary students’ life satisfaction in PISA 2018. Br J Educ Psychol. 2023;00: 1-25.

99. Prati, G. Correlates of quality of life, happiness and life satisfaction among European adults older than 50 years: A machine‐learning approach. ARCH GERONTOL GERIAT. 2022; 103: 104791.

2. After the previous arguments and after attending the recommendation, I recommend this paper to continue with the editorial process.

Response: We revised our reference list in order to ensure that it is complete and correct. We checked every single list one by one to check it is correct and it opens properly. We also checked the list to see we have cited papers that have been retracted, and we confirmed that there are no papers that have been retracted in the list. In the mean time, we found some typos and misinformation from the list, so we revised them one by one as follows:

13. Csikszentmihalyi M, Hunter J. Happiness in everyday life: The uses of experience sampling. J. Happiness Stud. 2003; 4: 185-199.

49. Chen Q, Chou CY, Chen CC, Lin JW, Hsu CH. The effect of leisure involvement and leisure satisfaction on the well-being of pickleball players. Sustainability. 2022; 14(1): 152.

70. Larson R, Csikszentmihalyi M. Experiential Correlates of Time Alone in Adolescence. J. Pers. 1978; 46(4): 677-693.

72. Toker B, Kalıpçı MB. Happiness among tourism students: a study on the effect of demographic variables on happiness. Anatolia. 2022; 33(3): 299-309. 

83. Hakoköngäs E, Puhakka R. Happiness from nature? Adolescents’ conceptions of the relation between happiness and nature in Finland. Leis. Sci. 2023; 45(7): 665-683. 

Other changes

1. We revised all the references in the list to conform to your reference formatting style.

2. In fig1, we deleted the red underlines (i.e., Multicollinearlity, XGBoost, Light GBM).

3. We found a typo (dada) in fig3 so we revised it correctly (data).

4. One author(Seong Man Park)’s affiliation has been changed from Department of General English Education, College of Libera Arts to Department of English Language, College of Foreign Languages), so we revised it accordingly. 

Thank you so much for your kind and detailed comments. 

All the best,

Eui Jae Kim, Hyun Wook Kang, Seong Man Park

---

## [Decision Letter · Decision Letter 2]

25 Mar 2024

Determinants of the Happiness of Adolescents:

A Leisure Perspective

PONE-D-23-32346R2

Dear Dr. Park,

We’re pleased to inform you that your manuscript has been judged scientifically suitable for publication and will be formally accepted for publication once it meets all outstanding technical requirements.

Kind regards,

Henri Tilga, PhD

Academic Editor

PLOS ONE

Additional Editor Comments (optional):

Reviewers' comments:

Reviewer's Responses to Questions

**Comments to the Author**

1. If the authors have adequately addressed your comments raised in a previous round of review and you feel that this manuscript is now acceptable for publication, you may indicate that here to bypass the “Comments to the Author” section, enter your conflict of interest statement in the “Confidential to Editor” section, and submit your "Accept" recommendation.

Reviewer #1: All comments have been addressed

2. Is the manuscript technically sound, and do the data support the conclusions?

Reviewer #1: Yes

3. Has the statistical analysis been performed appropriately and rigorously? 

Reviewer #1: Yes

4. Have the authors made all data underlying the findings in their manuscript fully available?

Reviewer #1: Yes

5. Is the manuscript presented in an intelligible fashion and written in standard English?

Reviewer #1: Yes

6. Review Comments to the Author

Reviewer #1: After reviewing your last version I have no further comments

This paper should continue with the normal process

7. PLOS authors have the option to publish the peer review history of their article (what does this mean?). If published, this will include your full peer review and any attached files.

Reviewer #1: No

---

## [Editor Report · Acceptance letter]

1 Apr 2024

PONE-D-23-32346R2 

PLOS ONE

Dear Dr. Park, 

I'm pleased to inform you that your manuscript has been deemed suitable for publication in PLOS ONE. Congratulations! Your manuscript is now being handed over to our production team.

Kind regards, 

on behalf of

Dr. Henri Tilga 

Academic Editor

PLOS ONE